# Inter-nesting area use, migratory routes, and foraging grounds for hawksbill turtles (*Eretmochelys imbricata*) in the Western Caribbean

Stephen G. Dunbar[1,2,3]*, Daniel R. Evans[4], Lindsey R. Eggers[5], Quintin D. Bergman[6], Luis G. Fonseca[7], Frank V. Paladino[6,8], Lidia Salinas[2,3], Chelsea E. Durr[6,8]

1 Marine Research Group, Loma Linda University, Loma Linda, California, United States of America, 2 Protective Turtle Ecology Center for Training, Outreach, and Research, Inc. (ProTECTOR, Inc.), Loma Linda, California, United States of America, 3 Protective Turtle Ecology Center for Training, Outreach, and Research, Inc. (ProTECTOR - Honduras), Tegucigalpa, Honduras, 4 Sea Turtle Conservancy, Gainesville, Florida, United States of America, 5 Seattle Aquarium, Seattle, Washington, United States of America, 6 Department of Biology, Purdue University Fort Wayne, Fort Wayne, Indiana, United States of America, 7 Biocenosis Marina, Trinidad de Moravia, San José, Costa Rica, 8 The Leatherback Trust, Fort Wayne, Indiana, United States of America

* sdunbar@llu.edu

## Abstract

The hawksbill turtle, *Eretmochelys imbricata*, has been at risk of extinction for more than 40 years and remains critically endangered. While nesting beach protection is important for hatchling production, identifying inter-nesting, migratory, and foraging habitats is crucial for mitigating threats to population recovery. We report the use of satellite telemetry to monitor movements of 15 hawksbill turtles in the Western Caribbean. Transmitters were deployed on nesting turtles in Honduras (2012 n = 2; 2017 n = 3), Costa Rica (2000 n = 2; 2014 n = 1; 2015 n = 1; 2018 n = 4; 2021 n = 1), and Panama (2017 n = 1). Hawksbill inter-nesting habitats ranged from 4-2,643 km² (core 50% utilization distribution) for the 15–70 tracking days. Large inter-nesting area use may be a result of habitats adjacent to a narrow continental shelf with strong ocean currents, causing turtles to actively search for suitable habitats. Following nesting, these turtles engaged in migrations to foraging grounds that covered 73–1,059 km lasting between 5–45 days. During migrations, turtles regularly altered their direction relative to ocean currents, using with-current movement to counteract against-current movement. Hawksbills from multiple beaches congregated in the same foraging habitat, despite nesting in different years. Turtles in this study foraged along the coastal and continental shelves of Nicaragua, Honduras, Belize, and Mexico, with turtles from disparate nesting sites utilizing the Nicaragua Rise hotspot area. Foraging area use was generally smaller (n = 8, 6–705 km²) than inter-nesting area use, possibly indicating that foraging habitats provided necessary food and resting areas. These data help us better understand inter-nesting and foraging habitat locations, core area use, and post-nesting migrations. Together, this provides vital information to mitigate potential in-water threats to critically endangered adult hawksbills along Western Caribbean migration corridors.

**Data availability statement:** The updated and publicly open data repository link is: https://purr.purdue.edu/publications/4740/1.

**Funding:** Funding for this project was provided by the Boyd Lyon Sea Turtle Fund to QDB, the Sonoma County Community Foundation to CED, the Fort Wayne Children's Zoo Conservation Fund and the Jack Schrey Distinguished Professor Funds to FVP, the USFWS Marine Turtle Conservation Fund Grant #611510 under the direction of Earl Possardt to SGD, the Department of Earth and Biological Sciences at Loma Linda University to SGD, as well as the California Turtle and Tortoise Club Inland Empire Chapter to SGD.

**Competing interests:** The authors have declared that no competing interests exist.

## Introduction

The hawksbill sea turtle, *Eretmochelys imbricata*, can be found in tropical waters around the globe [1]. This species has been exploited by humans for centuries for its meat and eggs, although direct take was primarily due to its elaborately colored carapace which is rendered into tortoiseshell products and sold internationally [2–5]. Trade in tortoiseshell was and is a primary cause of the severe decline in hawksbill populations worldwide [5,6]. Due to substantial decreases in numbers, the hawksbill turtle is currently listed as Critically Endangered on the International Union for Conservation of Nature (IUCN) *Red List of Threatened Species* [7]. While addressing the illegal tortoiseshell trade and implementing nesting beach protections are vital to conservation efforts for the species, these alone are insufficient to restore populations. A holistic approach to conservation, in which every life stage is addressed, is necessary if sea turtle populations are to recover [8–11]. Many sea turtle species, including hawksbills, migrate between foraging grounds and nesting sites, with studies indicating sea turtles may travel along established migratory corridors between the two. For example, Pendoley et al. [12] found fidelity to migrating corridors in flatbacks (*Natator depressus*) off the northwest coast of Australia, while the same behavior was reported by Marcovaldi et al. [13] in migrating loggerheads (*Caretta caretta*) in waters off northwestern Brazil. The existence of migratory corridors for specific turtle populations provides opportunities for protected species managers to take steps to protect turtles at both national and multinational scales.

Of the seven extant sea turtle species, the migration and movement patterns of hawksbills are the least well known [14]. Initially, researchers suggested that hawksbills did not embark on extensive migrations [1,15,16]. We now understand through both genetic analyses and satellite telemetry, that nesting hawksbills at a given site may aggregate from foraging grounds located in many different countries. For example, Meylan [17] collected tag return data on hawksbills in the Caribbean indicating that turtles were migrating from their nesting grounds in Tortuguero, Costa Rica, to a variety of countries, including Nicaragua, Honduras, and Panama. Troëng et al. [18] utilized flipper tag returns as well as satellite telemetry to document hawksbill migrations from Tortuguero, Costa Rica, to waters off Nicaragua and Honduras, including the Honduran island of Guanaja in the Bay Islands. A study by van Dam et al. [19] followed hawksbills nesting on Mona Island, Puerto Rico, and documented their journeys to the French West Indies, Nicaragua, Honduras, Turks and Caicos, the U.S. Virgin Islands, and the British Virgin Islands. Turtles have been found more than 2,000 km from where they were tagged and up to 15 years after they were tagged, indicating that hawksbill turtles are capable of the same dispersal patterns as other sea turtle species [19,20]. Satellite telemetry has been invaluable in addressing gaps in our knowledge with regard to at-sea movements and behavior of sea turtles, and while the data is plentiful for some species, sample sizes remain small particularly for hawksbills [21–23].

Therefore, we used satellite telemetry to identify the movements and habitat use of hawksbill turtles in the Western Caribbean to expand our knowledge of this imperiled species. Historically, extensive nesting populations of hawksbills were present throughout the Caribbean [24]. The Bay Islands off the north coast of Honduras; Tortuguero, Costa Rica; and the Bocas del Toro Province of Panama are all areas that were originally productive hawksbill nesting areas [6,25–27]. In Costa Rica, hawksbill populations saw a more than 77% decline between the 1950s and the early 2000s [18]. Similarly, in Panama, by the 1990s best estimates indicated that the populations had declined by almost 98% [6]. Some aspects of these declines have been addressed by the standard terrestrial and coastal protections. For example, Costa Rica has 166 protected areas that encompass 50% of the country's coastline, 20 of which are Marine Protected Areas (MPA) [28]. One of these is Tortuguero National Park, established in 1975 to

protect nesting turtles, nesting beaches, and adjacent terrestrial habitats. Another is an MPA that is incorporated in the Gandoca-Manzanillo Wildlife Refuge (GMNWR) which is included in the Area Conservación de la Amistad Caribe, located in southeast Costa Rica. Hawksbills are known to make nesting visits to the larger beach, Playa Gandoca, and the smaller beach, Playita, both of which are located within GMNWR [29]. However, this MPA was designed without specific knowledge of hawksbill spatial ecology; therefore, the GMNWR may lack sufficient protection of hawksbill habitats. While these coastal protections have varying degrees of success (i.e., more if they were informed by sea turtle research and potentially less if they were not), there is also need to address protection of migration routes and foraging habitats. These are more challenging because they may not be within any one country, existing, instead in international waters [30]. Known foraging habitats in the Western Caribbean are located along the continental shelves of Nicaragua and Honduras, as well as Belize and Mexico [14,18,30–35]. This is because hawksbill foraging requires coastal waters where potential prey items such as sponges, tunicates, corals, or algae [36–39] occur in sufficient abundance. These habitats provide important opportunities, yet require significant understanding of turtle use, in order to implement important conservation measures. Nevertheless, many threats to hawksbill turtles remain in the form of fisheries bycatch, direct intentional take of turtles in-water and at nesting beaches, and the loss of habitat due to degradation and coastal development with few regulatory restrictions.

In this study, we used satellite telemetry to 1) measure utilization distribution of area-restricted search behavior adjacent to nesting beaches (i.e., inter-nesting (IN) area); 2) define migratory corridors connecting nesting beaches to foraging grounds; 3) identify foraging habitats and quantify utilization distribution of area-restricted search behavior therein; and 4) calculate cosine similarity between direction of turtle movements in relation to ocean current directions.

## Methods

This study was conducted in strict adherence to ethical conditions for the treatment and care of animals, and in compliance with national laws and permitted research. For Honduras, ethical approval was provided by the Loma Linda University IACUC protocol approvals #89029 and #8150049 and Honduras national permits DGPA-005-2006 and DGPA-245-2006; for Costa Rica, Purdue University IACUC #1206000656 and permit numbers ACLAC-314-2018, ACLAC-315-2018, ACLAC-316-2018, ACLAC-317-2018, ACLAC-318-2018, and SINAC-ACTo-DIR-PI-RES-003-2024; for Panama, permit SE/A-56-2017.

### Nesting beach/transmitter deployment sites

We deployed satellite transmitters on 15 hawksbill sea turtles from four beaches in the Western Caribbean: Pumpkin Hill Beach (PHB), Utila, Honduras (PHB, Fig 1A); Tortuguero National Park, Costa Rica (TNP, Fig 1B); Gandoca-Manzanillo National Wildlife Refuge, Costa Rica (GMNWR, Fig 1C); and Chiriqui Beach, Panama (CBP; Fig 1D).

Utila is the smallest of the three Bay Islands in northern Honduras and is located approximately 35 km off the mainland. The Bay Islands lie at the southern end of the Mesoamerican Barrier Reef System, which extends north along the coasts of Guatemala, Belize, and Mexico, terminating at the tip of the Yucatan Peninsula. The island of Utila is 11 km long and 5 km wide, with two-thirds of the island covered by swamp and mangrove forests. Fourteen beaches (including nearby cays) have been assessed as potential nesting beaches for hawksbills [40], yet only two of these beaches have documented regular nesting. One is PHB (16° 07' N, 86° 53' W), where this study took place during the 2012 and 2017 nesting seasons, in

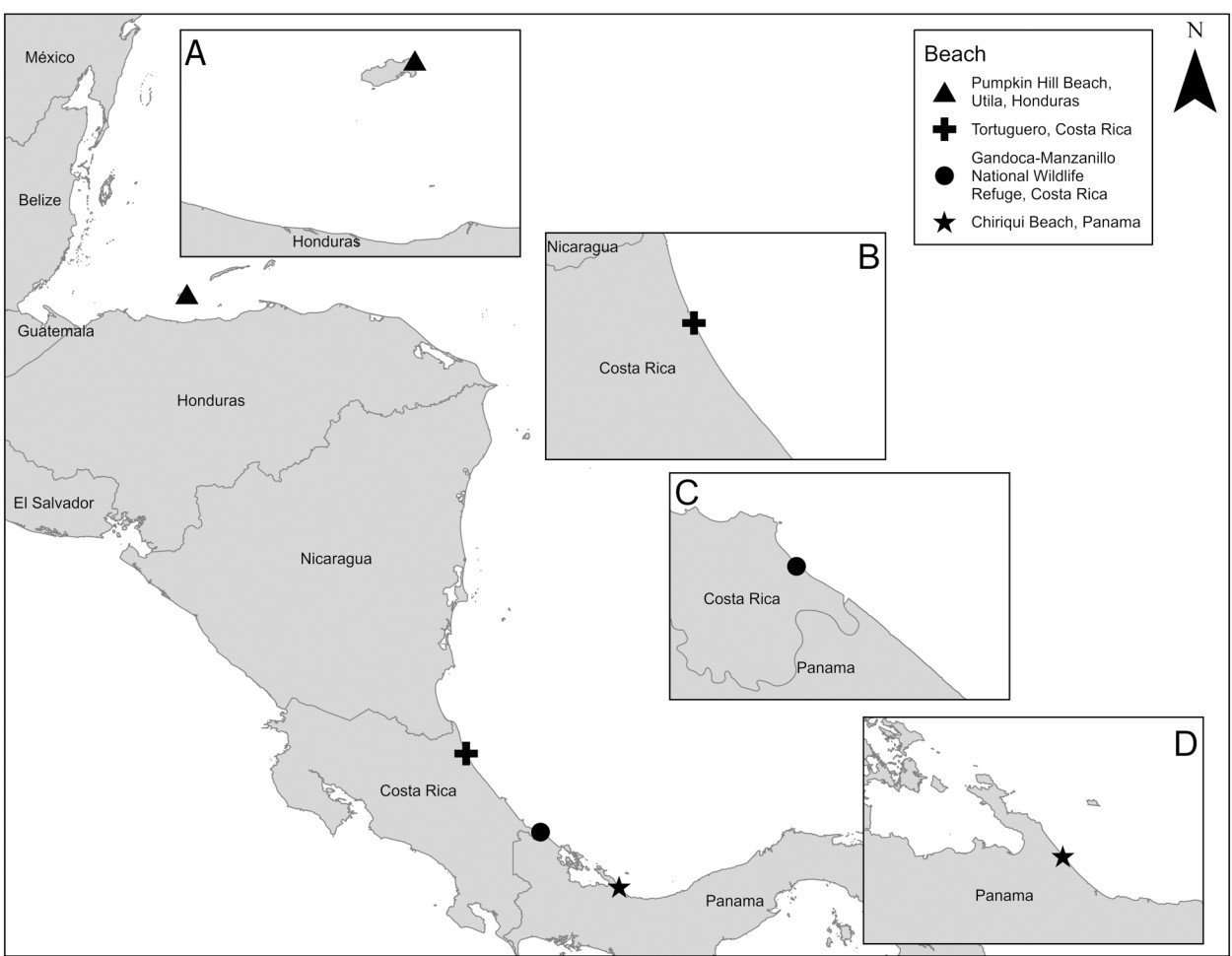

**Fig 1. Study site map including the locations of the beaches where we deployed satellite transmitters on nesting hawksbill sea turtles**
(*Eretmochelys imbricata* We deployed transmiters on A) Utila, Honduras in 2012 (n = 2) and 2017 (n = 3); B) Tortuguero, Costa Rica in 2000 (n = 2), 2014 (n = 1), 2015 (n = 1), and 2021 (n = 1); C) Gandoca-Manzanillo National Wildlife Refuge, Costa Rica in 2018 (n = 4); and D) Chiriqui Beach, Panama in 2017 (n = 1). Map was made in ArcGIS Pro using GADM shapefiles (https://gadm.org/).

conjunction with annual beach monitoring conducted jointly by ProTECTOR Inc., and the Bay Islands Conservation Association (BICA) Utila [41]. During the months of June through October, surveillance procedures involved walking the 1 km long beach nightly from 18:00 to 04:30 hours. We walked the length of the beach once per hour, as Stapleton and Eckert [42] determined that hourly patrols are likely to encounter almost all nesting turtles. Two nesting hawksbills in 2012 and three nesting hawksbills in 2017 encountered during beach monitoring were selected for platform terminal transmitter (PTT) application.

Tortuguero National Park, Costa Rica (10° 33' N, 83° 30' W) was established in 1975 to protect the largest green turtle rookery in the Western Hemisphere. The park includes over 19,000 hectares (46,900 acres) and protects 33.8 km of black sand nesting beach from the mouth of the Tortuguero River south to the river mouth at Jalova, just north of Parismina, Costa Rica. In addition to green turtles (*Chelonia mydas*), both leatherback (*Dermochelys coriacea*) and hawksbill turtles also nest in TNP. During the months of June through October, the northern 8 km of beach within the TNP were patrolled by the Sea Turtle Conservancy on foot nightly from 20:00 to 04:00 hours. The beach was patrolled each night during the entire time period to

encounter nesting turtles. We flipper-tagged nesting turtles, assessed turtle health conditions, recorded carapace measurements, and counted the number of eggs deposited. Two nesting hawksbills in 2000, one in 2014, one in 2015, and one in 2021 were encountered during beach monitoring and selected for PTT application.

Gandoca-Manzanillo National Wildlife Refuge, Costa Rica (09° 37' N, 82° 40' W) consists of 50.13 km² of terrestrial area and 44.36 km² of marine area for a total of 94.49 km² of protected habitat [43]. This includes some 10 km of beach which annually supports sea turtle nesting. Listed as a Ramsar site by the RAMSAR Convention on Wetlands of International Importance, GMNWR is a coastal lagoon consisting of seagrass beds, coral reefs, beaches, and cliffs with flooded lowland areas [44]. Anthropogenic uses in the area include traditional, low-scale agriculture growing cereals, cacao, plantains, yucca, and other tuberous plants; forestry; and marine and freshwater fishing [44]. Large-scale banana cultivation also occurs in the area adjacent to the reserve and communities (Q.D.B. personal observation). Created to protect endangered species and to maintain them in their natural habitat, GMNWR supports a high diversity of species including birds, reptiles, mollusks, fish (marine, estuarine, and freshwater), crustaceans (including lobsters), and 32 coral species [44]. Four of the five sea turtles of the Caribbean nest in GMNWR including the loggerhead (*Caretta caretta*), green, leatherback, and hawksbill turtles [29,45]. Four turtles were selected in 2018 for PTT attachment.

Chiriquí Beach, Panama (08° 56' N, 81° 39' W), in the Comarca Ngöbe-Buglé, is an important Caribbean hawksbill turtle rookery, as well as the most important nesting beach for leatherbacks in Caribbean Central America. Chiriquí Beach represents 22 km of beach along the Caribbean Coast of Panama. Starting in 2003, Sea Turtle Conservancy began a sustained presence monitoring Chiriquí Beach. Night patrols were conducted during the months of March through September for 6 hours each night to observe and flipper tag females. In addition, we collected data for each turtle including time, activity at first encounter, carapace measurements, presence of tag scars or overgrown tags, and any mutilations or deformities. One nesting hawksbill in 2017 was selected for PTT application.

## Study animals

All of the turtles tracked during this study were nesting female hawksbills with no apparent deformities (e.g., missing limbs, carapace malformations). Prior to transmitter deployment, we tagged the turtle with a unique identifying tag; either a Passive Integrated Transponder (PIT) tag (GMNWR) and/or metal Inconel 651-style tags, if one was not already present (PHB, TNP, CBP). When used, PIT tags were injected into the shoulder just under the skin, whereas metal Inconel tags were attached to the right front and/or right rear flippers, with tags placed on the proximal scale on the trailing edge of the flippers as per Dunbar and Berube [46] and Damazo [41]. Finally, we measured curved carapace length (CCL) and curved carapace width (CCW).

## PTT attachment

All transmitters were carapace-mounted transmitters generally following procedures from other transmitter studies [19,47,48] (Table 1). We cleaned the carapace using a combination of water, steel wool, sponges, sandpaper, and then isopropyl alcohol. The final step involved cross-hatching/scoring the second central scute to help the fastener adhere to the shell. At PHB, SPOT 5 and SPOT 293A (Wildlife Computers, Redmond, WA, USA) transmitters were attached using Sika Anchorfix two-part epoxy. At TNP, the ST-23 transmitters (Telonics, Mesa, AZ, USA) were attached using fiberglass cloth and resin with an added roll of Kevlar fiberglass anterior to the base of the antenna as additional protection for the antenna. The KiwiSat 202

Table 1. Data on transmitter attachment location, date, transmitter type, and area restricted search (ARS) as defined by the switching state-space model. ARS location points were then used to calculate the area (km²) of the 50% and 95% utilization distribution (UD). PHB = Pumpkin Hill Beach, Utila, Honduras; TNP = Tortuguero National Park, Costa Rica; GMNWR = Gandoca-Manzanillo National Wildlife Refuge, Costa Rica; CBP = Chiriqui Beach, Panama.

| Deploy Location | Transmitter ID | Deployment Date | Transmitter Type | Inter-Nesting ARS (d) | Inter-Nesting Area (km²) | | Foraging ARS (d) | Foraging Area (km²) | | Total Tracking (d) |
|---|---|---|---|---|---|---|---|---|---|---|
| | | | | | 50% UD | 95% UD | | 50% UD UD | 95% UD | |
| PHB | 108759 | Jul 11, 2012 | Wildlife Computers SPOT | 15 | 53 | 211 | 63 | 31 | 155 | 90 |
| | 108760 | Aug 12, 2012 | | – | – | – | 15 | 8 | 37 | 22 |
| | 119995 | Sep 13, 2017 | | – | – | – | 59 | 6 | 41 | 63 |
| | 120378 | Sep 13, 2017 | | 5* | 26 | 98 | 57 | 441 | 1,711 | 67 |
| | 138816 | Sep 14, 2017 | | 5* | 47 | 233 | 54 | 7 | 37 | 66 |
| TNP | 22126 | Jul 25, 2000 | Telonics ST-23 | 24 | 1,234 | 6,697 | 97 | 128 | 723 | 147 |
| | 22134 | Jul 26, 2000 | | – | – | – | 436 | 151 | 863 | 452 |
| | 137659 | Jul 4, 2014 | SirTrack KiwiSat 202 | 47 | 664 | 2,360 | 598 | 14 | 99 | 748 |
| | | | | 68 | 254 | 1,055 | – | – | – | – |
| | 129347 | Jul 2, 2015 | Wildlife Computers SPOT | 70 | 888 | 3,852 | 20 | 137 | 918 | 105 |
| | 220791 | Jul 13, 2021 | Wildlife Computers SPLASH 10 | 21 | 2,643 | 12,315 | 41 | 705 | 2,648 | 79 |
| GMNWR | 107904 | Aug 24, 2018 | SirTrack KiwiSat 202 | 55 | 201 | 1,170 | 276 | 242 | 1,972 | 355 |
| | 107915 | Aug 26, 2018 | | 12 | 17 | 86 | 784 | 23 | 279 | 1,254 |
| | | | | – | – | – | – | 414 | 27 | 266 |
| | 107910 | Aug 28, 2018 | | 42 | 213 | 1,301 | 315 | 15 | 145 | 370 |
| | 107913 | Aug 29, 2018 | | – | – | – | 472 | 28 | 353 | 517 |
| CBP | 160923 | May 30, 2017 | SirTrack KiwiSat 202 | 65 | 4 | 24 | 472 | 6 | 61 | 564 |

*These turtles did not display inter-nesting behavior because 5 days is not sufficient to nest a subsequent time. These represent the days the turtle spent in the vicinity of the nesting beach prior to migrating.

(Sirtrack/Lotek, Seattle, WA, USA) and SPOT 5 tags were attached using a two-part epoxy (Powers 308+, Powers Fasteners, Brewster, NY). Finally, the SPLASH10 was attached using the Wildlife Computers Sea Turtle Attachment Method [49] consisting of an epoxy putty, resin, and fiberglass. At GMNWR and CBP, KiwiSat 202 transmitters were attached using a two-part Pure50+ epoxy (Powers Fasteners, Brewster, NY), and two-part powers 308+ epoxy (Powers Fasteners, Brewster, NY), respectively. In all cases, this process took between 30 and 120 minutes and turtles were immediately permitted to return to the water, except at TNP, where turtles were kept in boxes overnight to have transmitters attached in the morning.

## Data analysis

We downloaded location data relayed via the Argos satellite system and removed any points that were on land. We then applied the Freitas et al. [50] location filter (sdafilter, argosfilter package) in R (R statistical software, R Version 4.3.1, Vienna, Austria). We removed locations requiring a max velocity of > 1.389 m/s and locations requiring a turn angle < 12° with a corresponding spike distance of 2,500 m or those requiring < 25° with a corresponding spike distance of 5,000 m [51–53]. Subsequently, a Bayesian state-space model was fitted to each set of filtered turtle location data individually using the "bsam" package in R [54–56]. Here, we used the switching correlated random walk model which estimates both location and behavioral states. Parameters for our model were a time step of one day, 10,000 Markov Chain Monte Carlo samples for adaptation and burn-in, 5,000 samples generated after burn-in, every 10th sample was retained to reduce autocorrelation, and a smoothing value of 0.3 was used.

This model returned a behavioral mode of either 1 or 2, where 1 is considered transitioning (migrating) behavior and 2 is considered area restricted search (ARS) behavior, which is usually associated with either inter-nesting (when ARS is located adjacent to the turtle's nesting beach) or foraging behavior (when ARS is located in oceanic habitats or those removed from known nesting beaches). This divided our data into two behavioral states. We used the "adehabitathr" package in R [57,58] to estimate kernel home range with 50% and 95% utilization distribution (UD) contours to represent core and resident areas, respectively, and generate shapefiles. Finally, for each track segment, defined as the straight-line distance between two consecutive location points (A and B), we calculated the direction. To do this, we used the Haversine equation to estimate the track length between the two points (A and B) and the east-west distance:

$$a = \sin^2\left(\frac{\varphi_B - \varphi_A}{2}\right) + \cos\varphi A \cdot \cos\varphi B \cdot \sin^2\left(\frac{\lambda_B - \lambda_A}{2}\right)$$

$$c = 2 \cdot atan2\left(\sqrt{a},\ \sqrt{1-a}\right)$$

$$distance = R \cdot c$$

where $\varphi$ is the latitude, $\lambda$ is the longitude, atan2 measures the counterclockwise angle $\theta$, in radians, between the positive x-axis and the point, and R is earth's radius (mean radius = 6,371 km). Together these segments allowed us to calculate the angle from due East.

In ArcGIS Pro (Version 3.2.1, 2023, Esri, Redlands, CA, USA), we mapped individual tracks, and calculated UD area in ArcGIS from shapefiles and spatial data (country shapefiles: https://gadm.org/; bathymetry shapefile: https://www.gebco.net/; and economic exclusion zones shapefile: www.marineregions.org). We downloaded temporally relevant historical satellite data for this project due to the historical nature of the turtle movements. Ocean current direction was calculated by using data from Copernicus Marine Services (https://marine.copernicus.eu/) and ArcGIS Pro to transform vector information into the angle from due East. In this way, we could calculate the cosine similarity $S_C$ between the two angles using:

$$S_C\left(\theta_1, \theta_2\right) = \cos\left(\theta_1 - \theta_2\right)$$

where $\theta_1$ and $\theta_2$ are angles, calculated from due East, for the hawksbill migration track and ocean current movement at the track segment, respectively. An angle close to 0° or 360° indicates a direction of movement that is east, while an angle of approximately 180° indicates a westward movement direction. An $S_C$ of $-1$ means the movement direction of the turtle is directly against the direction of ocean current movement. An $S_C$ of 1 indicates that the turtle is moving in the same direction as the ocean currents. We then weighted the $S_C$ by the track length. All results are presented as mean ± 1 standard deviation.

## Results

### Inter-nesting and nest adjacent ARS

We deployed satellite transmitters on 15 hawksbill turtles between 2000 and 2021 (Table 1). Turtles were tracked between 22 and 1,254 days, spending between 0 and 70 IN days in the vicinity of the nesting beach before engaging in post-nesting migrations.

In Honduras, five turtles were tracked from PHB. Post-tagging, turtle 108759 engaged in inter-nesting behavior for 15 days and had a core UD of 53 km² before nesting again (Table 1). Turtles 120378 and 138816 each spent 5 days in the area of Utila (core UDs of 26 and 47 km², respectively) before engaging in post-nesting migrations (Fig 2A). For these three turtles, ARS was associated with the southern side of the island where water was less than 100 m deep. The final two turtles, 108760 and 119995, began their post-nesting migrations immediately.

At TNP, turtles 22126, 137659, 129347, and 220791 engaged in inter-nesting ARS from between 21 to 70 days, suggesting that each of these turtles likely nested at least one subsequent time, in some cases possibly more (Fig 2B). Turtle 22126 engaged in inter-nesting behavior for 24 days and had a core area of 1,234 km² (Table 1). For turtle 137659, we recorded two inter-nesting seasons, one in 2014 that lasted 47 days (50% UD: 664 km²) when the transmitter was deployed and one in 2016 (50% UD: 254 km²) after re-migration, which lasted 68 days until transmissions stopped. Turtle 129347 spent 70 inter-nesting days in a core area of 888 km², and turtle 220791 remained near Tortuguero for 21 days in a core area of 2,643 km² (Table 1, Fig 2B). Overall, core IN area use was 1,357 ± 889 km².

At the GMNWR, Costa Rica, four hawksbill turtles were tracked starting in August 2018. Three turtles (107904, 107915, 107910) engaged in inter-nesting behavior that lasted between 13, 28, and 56 days and had core UDs of 17, 213, and 201 km², respectively (Table 1, Fig 2C). Turtles 107904 and 107915 remained near the nesting beach, while turtle 107910's IN duration (42 d) was split between the nesting beach area and Bocas del Toro, Panama. It is likely that these turtles each nested a subsequent time(s). Taken together, turtles at GMNWR had a mean core IN area of 144 ± 110 km².

At CBP, a single turtle (160923) was tagged in 2021. This turtle spent 65 d in a coastal inter-nesting habitat north of the nesting beach, with this amount of time sufficient to nest multiple times (Table 1, Fig 2D). Core UD was 4 km² and residential UD was 24 km² (Table 1).

## Migration

Migrations lasted 5 to 46 days (mean = 22 ± 12 d SD) and turtles traveled between 73 and 1,522 km (mean = 585 ± 433 km SD), remaining south of the Caribbean Current. Turtles displayed an average calculated migration travel speed of 26 ± 9.5 km/d (7–43 km/d), and preferred coastal and continental shelf foraging in Nicaragua, Honduras, Belize, and Mexico where water was shallow and water currents were slower (Fig 3).

Turtles nesting on PHB engaged in migrations that lasted between 5 and 33 days and covered between 73 and 434 km track distance (Table 2). Turtle 108759 began her post-nesting migration by traveling from Utila in a northwesterly direction, reaching the mainland of Belize at Ambergris Caye (Fig 4). From here, the turtle traveled north along the coast until reaching a foraging habitat in Bahia de la Ascension, Mexico (Fig 4). The second turtle tagged in August 2012 (108760) departed the nesting area immediately, and traveled 202 km in 8 days to reach Middle Long Caye, Belize (Table 2, Fig 4). Of the three turtles tagged in 2017, 119995 departed directly, while turtles 120378 and 138816 spent 5 days around Utila before migrating. Turtle 119995 traveled 73 km in 5 days and began foraging approximately 40 km off the west coast of Utila (Table 2, Fig 4). Turtle 120378 traveled 237 km in 33 days. During the initial 5 days, the turtle crossed over open ocean and skirted the south end of Glover's Reef, then headed southwest to the waters off of the city of Placencia. After this, the turtle engaged in a wandering style behavior through the cays, finally settling east of Laughing Bird Caye, Belize. This wandering behavior was included in both the migration map (Fig 4) and the foraging UD (Table 2). Finally, turtle 138816 traveled 243 km in 13 days. After 7 days, the turtle reached the cays at the south end of Glover's Reef Atoll at which point she migrated north in the sheltered

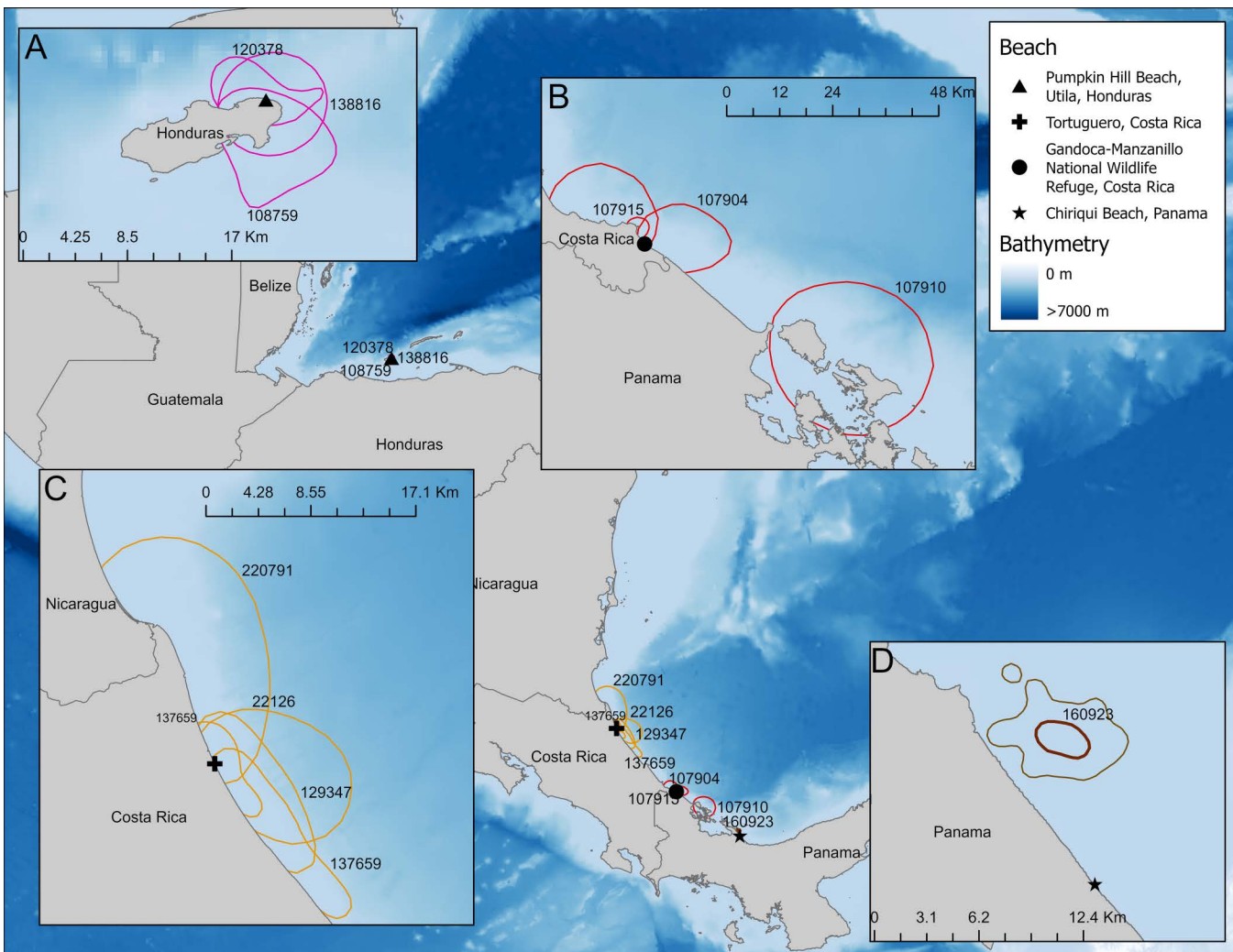

**Fig 2. Inter-nesting area restricted search for core (50%) utilization distribution of hawksbill sea turtles nesting on:** (A) Pumpkin Hill Beach, Utila, Honduras (pink); (B) Tortuguero, Costa Rica (orange); (C) Gandoca-Manzanillo National Wildlife Refuge, Costa Rica (red); and (D) **Chiriqui Beach, Panama (brown).** Map was made in ArcGIS Pro using GADM shapefiles (https://gadm.org/) and GEBCO bathymetry. For Chiriqui Beach, we included both core and resident (95%) contour lines.

water, ending between Drowned Cayes (Middle Long Caye)/Turneffe Atoll and mainland Belize (Belize City, Fig 4).

Turtles nesting at TNP engaged in coastal migratory routes traveling between 300 and 620 km over 16 to 29 days, reaching foraging grounds between central and northern Nicaragua (Table 2, Fig 5). Turtles 22126 and 22134 traveled 426 and 301 km, respectively, in 29 and 17 days (Table 2). Turtle 137659 migrated 620 km in 22 days to reach the continental shelf off Northern Nicaragua, then, in 2016, remigrated 465 km in 16 days to return to TNP (Fig 6). Turtle 129347, tagged in 2015, migrated 363 km ending on the continental shelf of central Nicaragua (Table 2, Fig 5). Turtle 220791 traveled 499 km in 19 days to forage on Half Moon Reef, on the continental shelf adjacent to the border of Nicaragua and Honduras (Table 2; Fig 5).

From GMNWR, turtle 107904 traveled 970 km in 26 days ending at Half Moon Reef (Table 2; Fig 7). Turtle 107915 traveled from GMNWR to Utila, Honduras to forage between Utila and the mainland, a migration that took 44 days and covered 1,522 km (Fig 7). We tracked turtle 107910

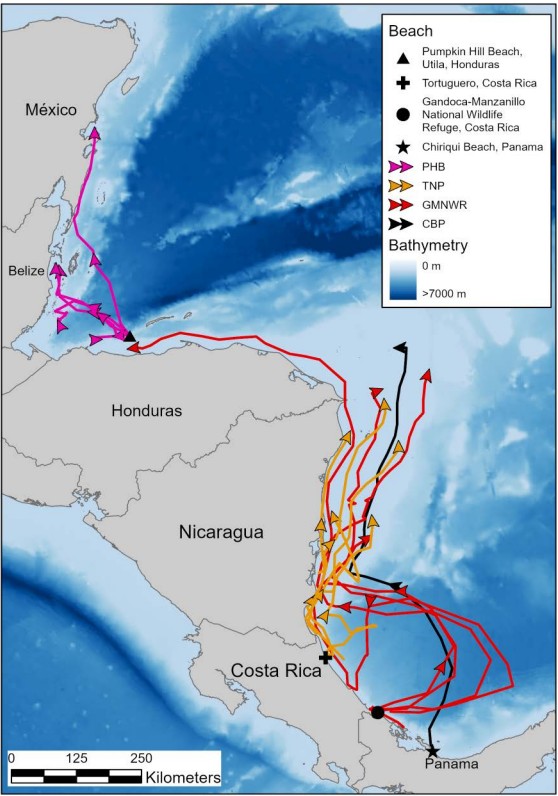

**Fig 3. Movements and area use of hawksbill turtles in relation to nesting beaches and bathymetry.** Turtle tracks in pink were tagged on Pumpkin Hill Beach, Utila, Honduras (PHB); orange tracks were from turtles tagged at Tortuguero, Costa Rica (TNP); red tracks were from turtles tagged on Gandoca-Manzanillo National Wildlife Refuge, Costa Rica (GMNWR); and the brown track was from a turtle tagged on Chiriqui Beach, Panama (CBP). Deployment dates can be found in Table 1. This map was made using bathymetry from GEBCO, and the map was made using ArcGIS Pro.

for 605 km over 14 days, where she reached a foraging habitat between Corn Island and San Andreas/San Luis, in central Nicaragua (Fig 7). Finally, turtle 107913 looped out away from the shore into the Colombia-Panama Gyre before returning to move along the coastline in central Costa Rica and continuing north to Half Moon Reef, a trip that took 46 days and covered 1,494 km (Table 2, Fig 7).

The single turtle that nested on CBP (160923) traveled 924 km in 30 days to forage alongside the Costa Rican hawksbills on the Nicaraguan Rise (Table 2; Fig 7).

## Foraging habitat ARS

The longest recorded foraging period was 1,198 days, and foraging time seemed to be habitat specific, with turtle 137659 returning to the nesting beach (remigration) after less than two years in her foraging habitat (598 d), and turtle 107915 remaining in her foraging habitat for more than 1,198 days. Turtle 107915 was recorded foraging in a primary foraging area for 784 days before moving to a second foraging area for 414 days until the transmitter failed.

Hawksbill turtles nesting at PHB used coastal foraging habitats within Honduras, Belize, and Mexico in areas that included reefs, shallow water, and sometimes coastal inlets (Fig 8). Core foraging ARS ranged from 6 km² to 441 km² (Table 2) and were generally smaller than that of inter-nesting core ARS (Table 1). Turtle 120378 engaged in coastal wandering which

**Table 2. Migration data for hawksbill turtles nesting in the Western Caribbean, including nesting beach/tagging location, migration dates, migration track length, average travel speed, average travel direction in degrees from due east, average ocean current direction in degrees from due east, and the average daily cosine similarity (weighted $S_C$) between the direction of turtle movement and the direction of water movement, weighted by the daily track length. Migration track lengths were calculated using the Haversine formula. PHB = Pumpkin Hill Beach, Utila, Honduras; TNP = Tortuguero National Park, Costa Rica; GMNWR = Gandoca-Manzanillo National Wildlife Refuge, Costa Rica; CBP = Chiriqui Beach, Panama.**

| Beach | Tag | Migration Dates | Migra-tion (d) | Track Length (km) | Speed (km/d) | Track Direction | Current Direction | Weighted Average $SC$ |
|---|---|---|---|---|---|---|---|---|
| PHB | 108759 | July 25–August 7, 2012 | 14 | 433.89 | 30.99 | 176.92 | 95.82 | 0.13 |
| | 108760 | August 13–August 20, 2012 | 8 | 201.97 | 25.25 | 121.27 | 112.77 | 0.46 |
| | 119995 | September 14–September 18, 2017 | 5 | 72.92 | 14.58 | 176.22 | 186.52 | -0.12 |
| | 120378 | September 18–October 20, 2017 | 33 | 237.46 | 7.20 | 191.93 | 168.90 | 0.06 |
| | 138816 | September 15–September 27, 2017 | 13 | 242.50 | 18.65 | 132.57 | 187.23 | 0.13 |
| TNP | 22126 | August 17–September 14, 2000 | 29 | 426.21 | 14.70 | 109.71 | 233.15 | -0.08 |
| | 22134 | July 29–August 14, 2000 | 17 | 300.62 | 17.68 | 103.51 | 213.52 | 0.05 |
| | 137659 | August 21–September 11, 2014 | 22 | 619.33 | 28.15 | 79.91 | 215.56 | 0.00 |
| | | May 1–May 15, 2016 | 16 | 465.02 | 29.06 | 258.10 | 167.11 | 0.09 |
| | 129347 | September 10 - September 26, 2015 | 17 | 363.37 | 21.37 | 117.12 | 185.20 | 0.10 |
| | 220791 | August 2–August 20, 2021 | 19 | 498.74 | 26.25 | 73.27 | 168.90 | 0.11 |
| GMNWR | 107904 | October 17–November 11, 2018 | 26 | 969.75 | 37.30 | 87.32 | 169.55 | 0.08 |
| | 107915 | September 6–October 17, 2018 | 44 | 1,521.69 | 34.58 | 138.37 | 185.46 | 0.09 |
| | 107910 | October 09–October 21, 2018 | 14 | 604.87 | 43.21 | 85.44 | 234.82 | -0.10 |
| | 107913 | August 29–October 13, 2018 | 46 | 1,493.98 | 32.48 | 111.58 | 183.55 | 0.05 |
| CBP | 160923 | August 3–September 1, 2017 | 30 | 923.57 | 30.79 | 93.05 | 172.73 | 0.08 |

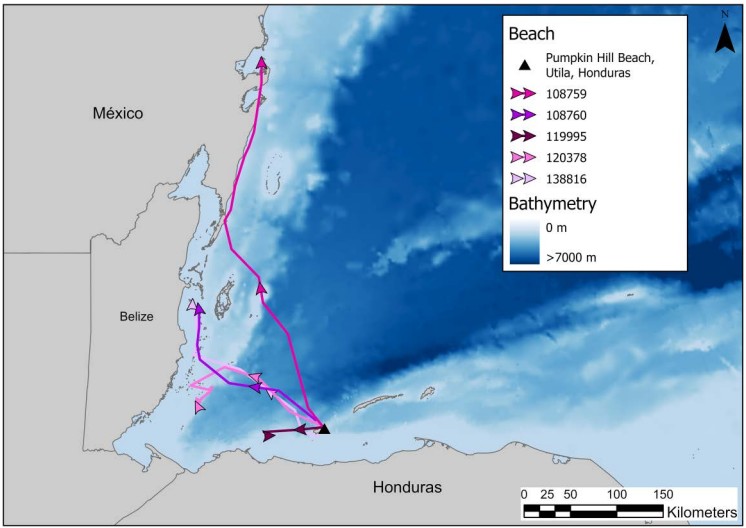

**Fig 4. Migrations of hawksbill sea turtles nesting on Pumpkin Hill Beach, Utila, Honduras.** Turtles 108759 and 108760 were tagged in 2012; turtles 120378, 138816, and 119995 were tagged in 2017. Map was made in ArcGIS Pro using GADM shapefiles (https://gadm.org/index.htm) and GEBCO bathymetry.

we included in the UD calculations, resulting in a foraging core use area larger than the other foraging UDs. We did not have long-term recording of foraging location data from the PHB turtles (< 63 d). Taken together, core foraging area for turtles nesting at PHB was 99 ± 192 km².

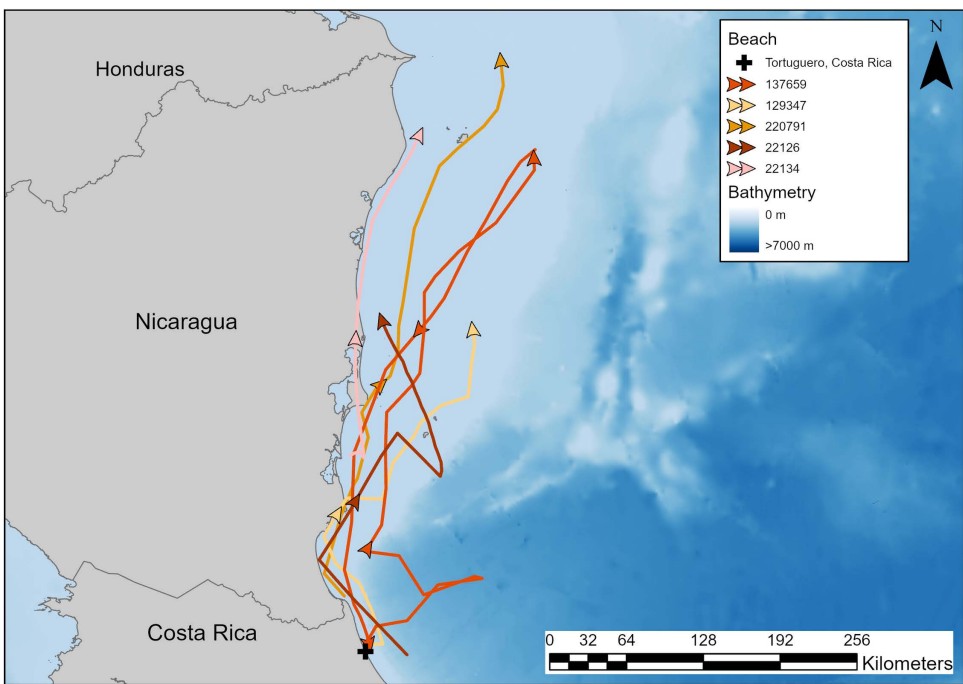

**Fig 5. Post-nesting migrations of hawksbill sea turtles nesting at Tortuguero, Costa Rica.** Turtles 22126 and 22134 were tagged in 2000, turtle 137659 was tagged in 2014, turtle 129347 was tagged in 2015, and turtle 220791 was tagged in 2021. Map was made in ArcGIS Pro using GADM shapefiles (https://gadm.org/) and GEBCO bathymetry.

Turtles tagged at TNP foraged along the Nicaraguan continental shelf (Fig 8). These turtles had core foraging UDs that ranged from 14 km² to 705 km² (Table 2). Each turtle had a smaller foraging UD compared to their respective inter-nesting UD. Turtle 137659 had the longest recorded number of foraging days (598 d) before migrating back to TNP to nest again. The core foraging area for this turtle (14 km²) was the smallest used by any of the turtles tracked from TNP. On average, core foraging area was 227 ± 273 km².

Turtles nesting at GMNWR foraged in the same habitat as the turtles from TNP (n = 3) and adjacent to PHB (n = 1; Fig 8). Turtles that foraged along the continental shelf of Nicaragua were tracked between 263 and 463 days (Table 1) and had core UDs of between 23 km² and 260 km² (Table 2). Turtle 107915 foraged south of Utila Island (Fig 8). For the first 784 days the turtle had a primary foraging core UD of 23 km², then moved west approximately 100 km and spent 414 days in a secondary foraging UD of 27 km² (Table 2). This turtle did not return to the initial area for the duration of signal transmissions. Overall, mean foraging area for turtles nesting at GMNWR was 84 ± 106 km².

Finally, the turtle tagged on CBP also foraged in the same habitat as turtles from TNP and GMNWR, along the Nicaragua Rise (Fig 8). This turtle was tracked for 472 days and had a very small core UD of 6 km² (Table 2).

## Influence of currents on turtle movements

Cosine similarity ($S_C$) measures the angle of the direction of the turtle's movement and the ocean current movement. In figures 9, 10, and 11, sections in green were designated as generally with the currents, sections in yellow were designated as generally across the currents, and sections in red were generally against ocean currents. Each panel represents the migration of a single turtle whose satellite tag number is provided on the left above each panel.

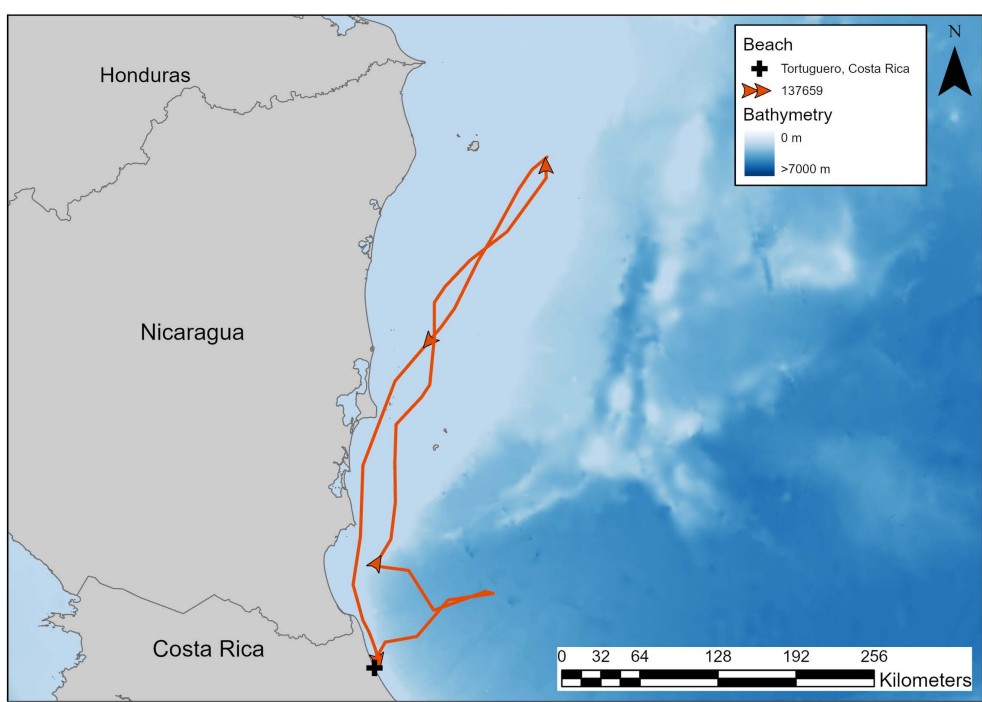

**Fig 6. Post-nesting (2014) and remigration (2016) of hawksbill turtle 137659 from nesting beach, Tortuguero, Costa Rica, to foraging habitat in northern Ni** Map was made in ArcGIS Pro using GADM shapefiles (https://gadm. org/) and GEBCO bathymetry.

Of the turtles nesting on PHB, turtle 108759 traveled against ocean currents between the island and the mainland of Belize, and then with ocean currents up the coast to Mexico. This led to an average weighted (by track segment length) $S_C$ between the track direction and the ocean current direction of 0.07 (Table 2, Fig 9). The with-current movement and against-current movement canceled each other out. The second turtle (108760) traveled predominantly against prevailing ocean currents (weighted $S_C$ of −0.33) to Middle Long Caye, Belize (Table 2). Turtle 119995 traveled perpendicular to ocean current movement and stopped off the west coast of Utila (weighted $S_C$ = 0.003; Table 2). Turtle 120378 zigzagged across ocean currents to Laughing Bird Caye in a way that averaged overall weighted $S_C$ (−0.01; Table 2). Finally, turtle 138816 traveled mostly perpendicular to ocean currents the entire time to the Drowned Cayes (weighted $S_C$ = 0.04; Table 2).

From TNP, turtle 22126's migration direction had a weighted $S_C$ = 0.06 compared to ocean current direction ending in central Nicaragua, while turtle 22134 had a weighted $S_C$ = 0.09 and ended in northern Nicaragua; both traveled mostly against the water current (Fig 10). Turtle 137659 migrated to the continental shelf off Northern Nicaragua, then re-migrated two years later to nest in TNP. The post-nesting migration swapped back and forth across the current, so that the turtle spent a few days in each direction (weighted $S_C$ = −0.02; Table 2). The remigration, however, was 8 days against the current with one day (day 6) traveling with the current until the turtle was able to catch a current and spend the last 7 days traveling with the currents (weighted $S_C$ = −0.21; Fig 10). Turtle 129347 migrated with water current direction for 12 out of 18 days (weighted $S_C$ = 0.05; Fig 10) to reach the continental shelf of central Nicaragua. Finally, turtle 220791 travelled across ocean currents (weighted $S_C$ = 0.04) to forage on Half Moon Reef, along the Nicaragua Rise on the continental shelf adjacent to the border of Nicaragua and Honduras.

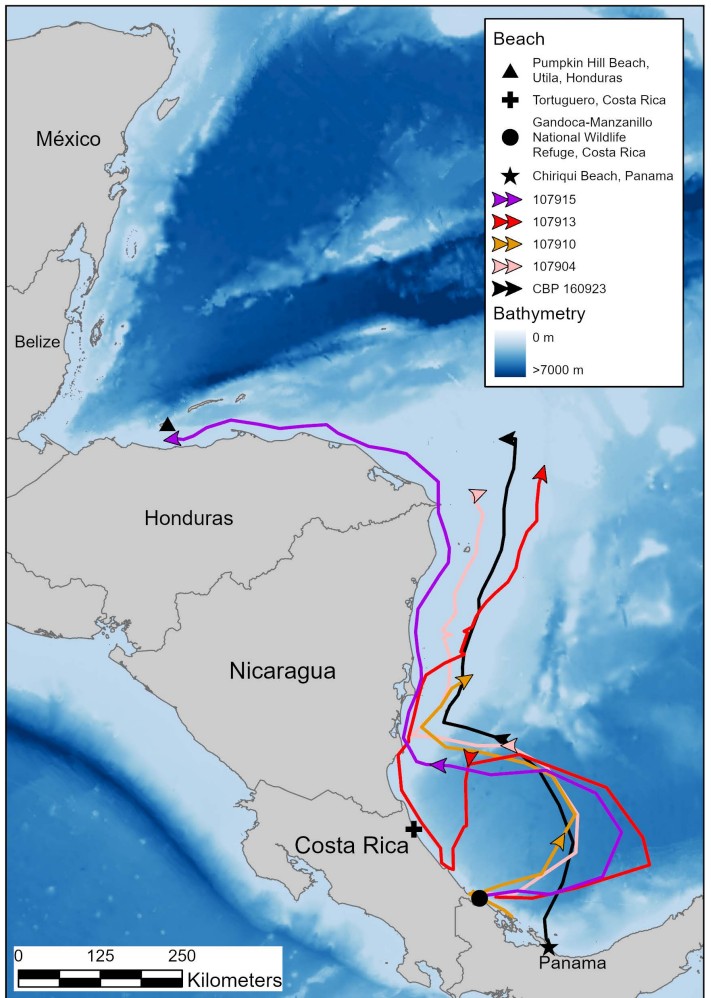

**Fig 7. Migrations of hawksbill sea turtles nesting on Gandoca-Manzanillo National Wildlife Refuge, Costa Rica, in 2018, and one turtle nesting on Chiriqui Beach, Panama in 2017.** Map was made in ArcGIS Pro using GADM shapefiles (https://gadm.org/) and GEBCO bathymetry.

At GMNWR, all of the turtles seemed to ride the Colombia-Panama Gyre out and then returned to the coast before turning north. Turtles 107904 and 107910 returned to the continental shelf in southern Nicaragua and then traveled north, ending at the Nicaraguan Rise (weighted $S_C$ = 0.08) and central Nicaragua continental shelf (weighted $S_C$ = −0.10), respectively, a journey that was mostly with the direction of the currents, although ending in central Nicaragua required a final against-current movement (Fig 11). Turtle 107915 also returned near shore around central Nicaragua, but continued around the horn of Nicaragua to Honduras, a journey that followed the direction of the local ocean currents (weighted $S_C$ = 0.09). Finally, turtle 107913 returned to the shore in central Costa Rica, requiring more days swimming against ocean currents before turning to move with the currents along the coast to the Nicaragua Rise and Half Moon Reef (weighted $S_C$ = 0.05).

From CBP, turtle 160923 traveled to the Nicaragua Rise and had an average weighted $S_C$ = 0.08, mostly travelling perpendicular and against the currents (Fig 11).

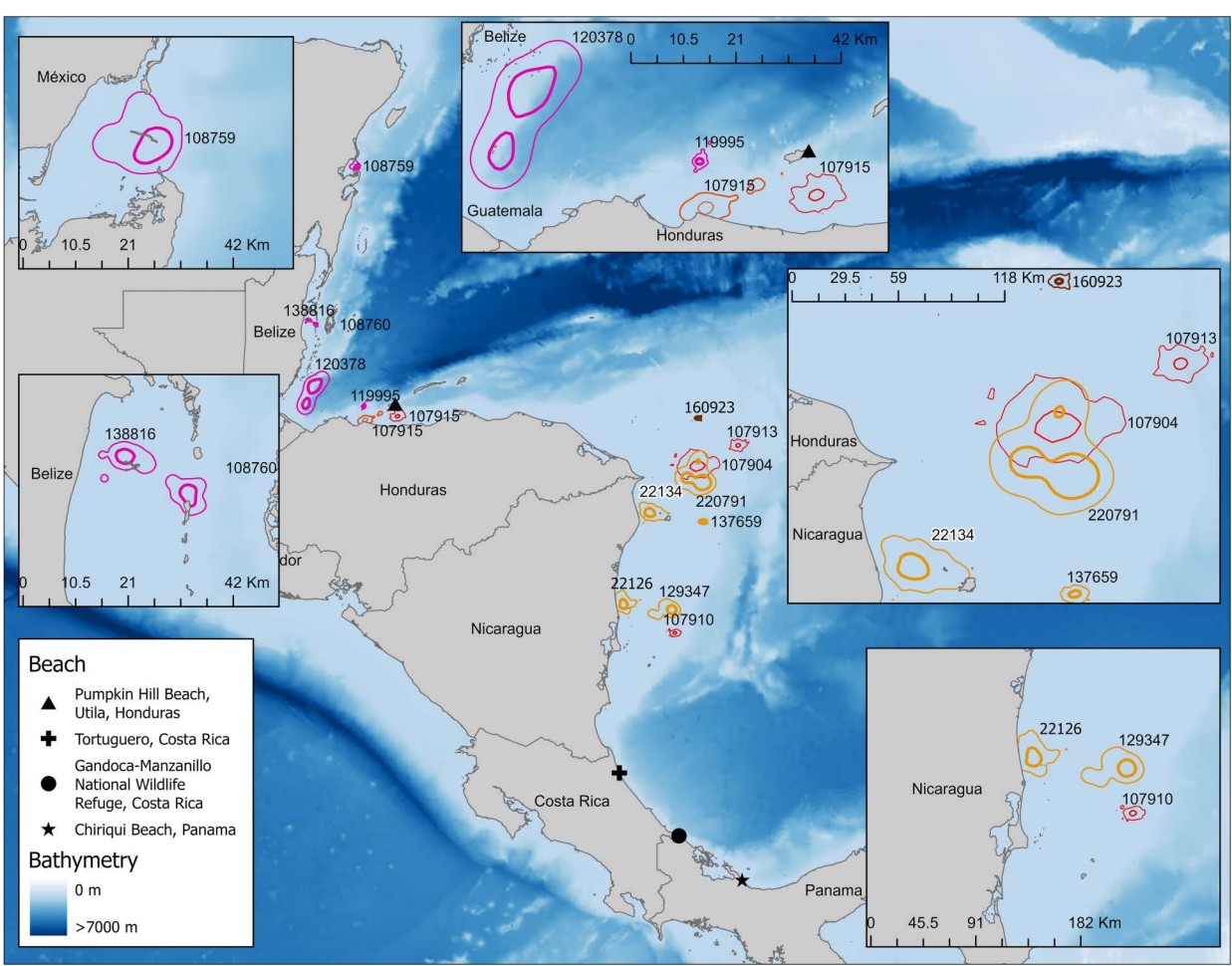

**Fig 8. Foraging area restricted search (95% UD with 50% UD bolded inset) behavior of hawksbill sea turtles nesting on Pumpkin Hill Beach, Utila, Honduras (pink); Tortuguero, Costa Rica (orange); Gandoca-Manzanillo National Wildlife Refuge, Costa Rica (red); and Chiriqui Beach, Panama (brown).** Map was made in ArcGIS Pro using GADM shapefiles (https://gadm.org/) and GEBCO bathymetry.

## Discussion

Although prior reports by Godley et al. [21] and Hays and Hawkes [23] recognized that most satellite telemetry had been accomplished on loggerheads (*C. caretta*), greens (*C. mydas*), and leatherbacks (*D. coriacea*), recent studies by Maurer et al. [32], and a compilation of some 258 tracks from 16 countries and overseas territories by Maurer et al. [30], have vastly increased our understanding of inter-nesting and post-nesting movements of hawksbills in the Western Atlantic over the last 10 years. Nevertheless, tracking data from the Central American countries of Belize, Honduras, Nicaragua, and Costa Rica are generally lacking in assessments of telemetry studies. Here, we present results that begin to fill gaps in our knowledge for hawksbills nesting in Honduras, Costa Rica, and Panama.

We used satellite telemetry to track hawksbill movements from four nesting beaches in three countries along the far western region of the Caribbean at different times between 2000–2021, in which IN area use was generally close to the nesting beaches and was larger than foraging area size. Migration paths zig-zagged across currents, possibly in order to limit against-current movement. Foraging areas were located along the continental shelves of Nicaragua, Honduras, Belize, and Mexico.

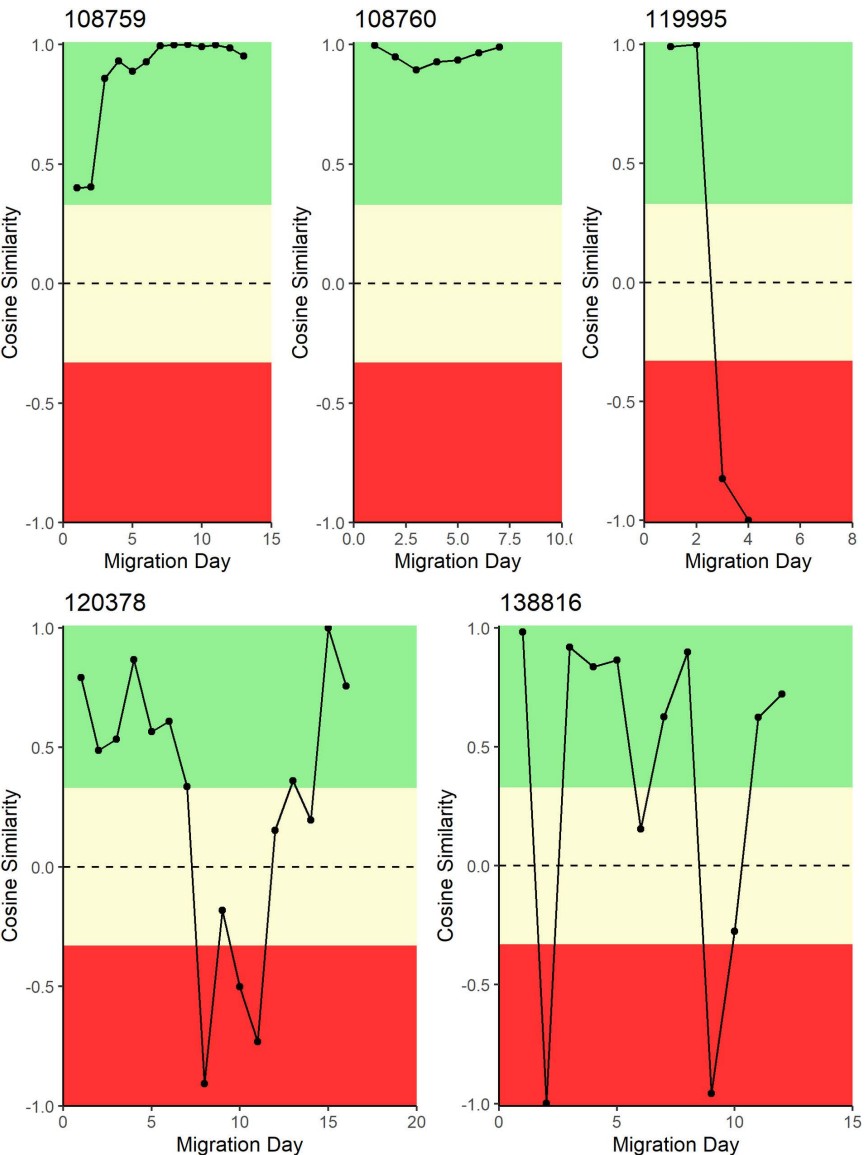

**Fig 9. Daily cosine similarity between turtle migration direction and ocean current direction for turtles tagged on Pumpkin Hill Beach, Utila, Honduras.** Cosine similarity close to +1 indicates that the direction of turtle movement was with ocean currents, whereas a cosine similarity close to −1 indicates a turtle movement direction against ocean currents.

## Inter-nesting

In this study, we observed a wide range of IN area use, from a core area use of 4 km² over a period of 65 days near CBP, to 2,643 km² over a period of 21 days off the coast of TNP. The continental shelf in Costa Rica is narrow, and area use indicates that turtles spent time offshore in deep water, adjacent to the Colombia-Panama Gyre. This current system may prevent the turtles from resting in smaller areas as there is less geographic protection compared to Honduras, which has shallow waters < 100 m deep, or Panama, where the continental shelf is shallow and allowed the entire residential UD to be well within the 50 m depth contour. Small IN area use estimated for PHB and CBP turtles are common for

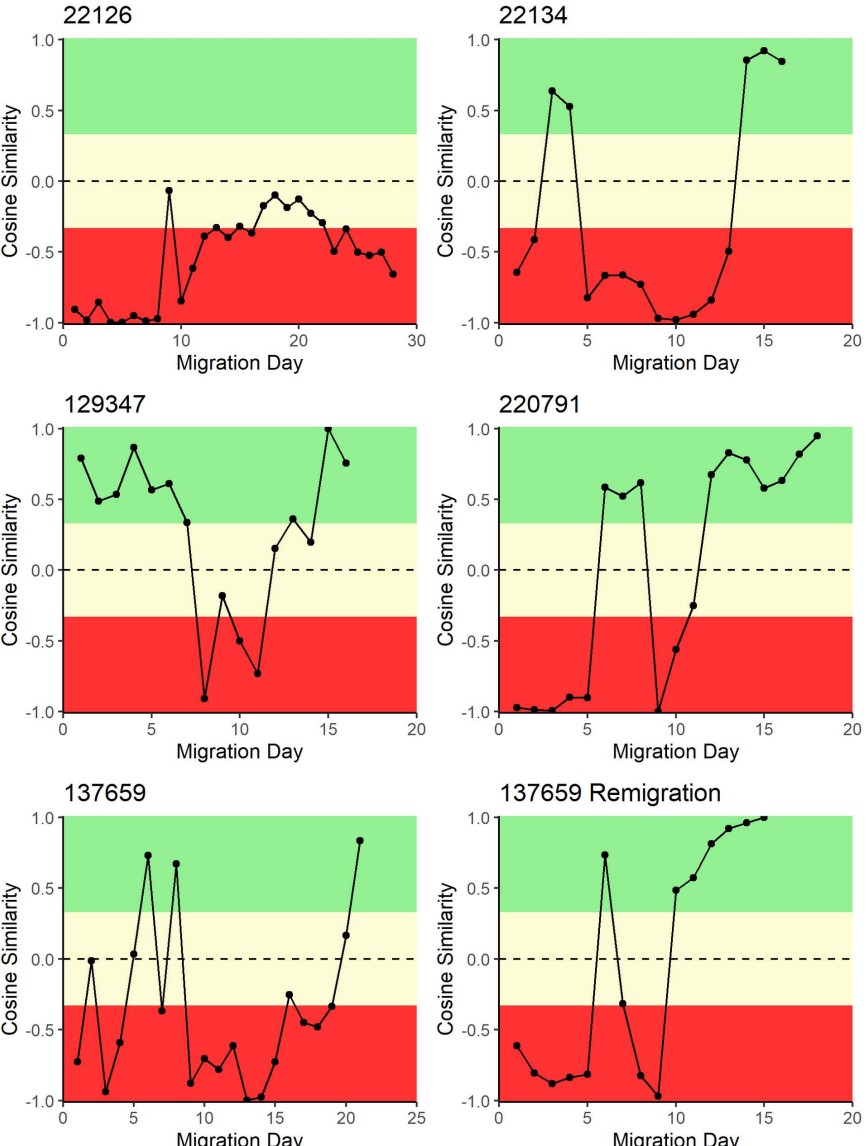

**Fig 10. Daily cosine similarity between turtle migration direction and ocean current direction for turtles tagged on Tortuguero, Costa Rica.** Cosine similarity close to +1 indicates that the direction of turtle movement was with ocean currents, whereas a cosine similarity close to −1 indicates a turtle movement direction against ocean currents. For turtle 137659, both a post-nesting migration and a remigration are presented.

hawksbill turtles in Caribbean islands, such as the Dominican Republic [33], St. Croix [35], Jamaica, and Antigua [32]. The IN area use for turtles at TNP were larger than those found at the other sites, suggesting that currents may pull turtles away from the nesting beach, creating an environment that potentially lacks sufficient resting areas resulting in more active IN periods. This is likely to have significant energetic costs for turtles, especially if they are fasting during the nesting season. For example, turtles nesting at GMNWR were all swept out into the Colombia-Panama Gyre upon beginning their migrations. If this were to happen to a turtle during IN attempts, it may potentially be prohibitive to subsequent nesting efforts.

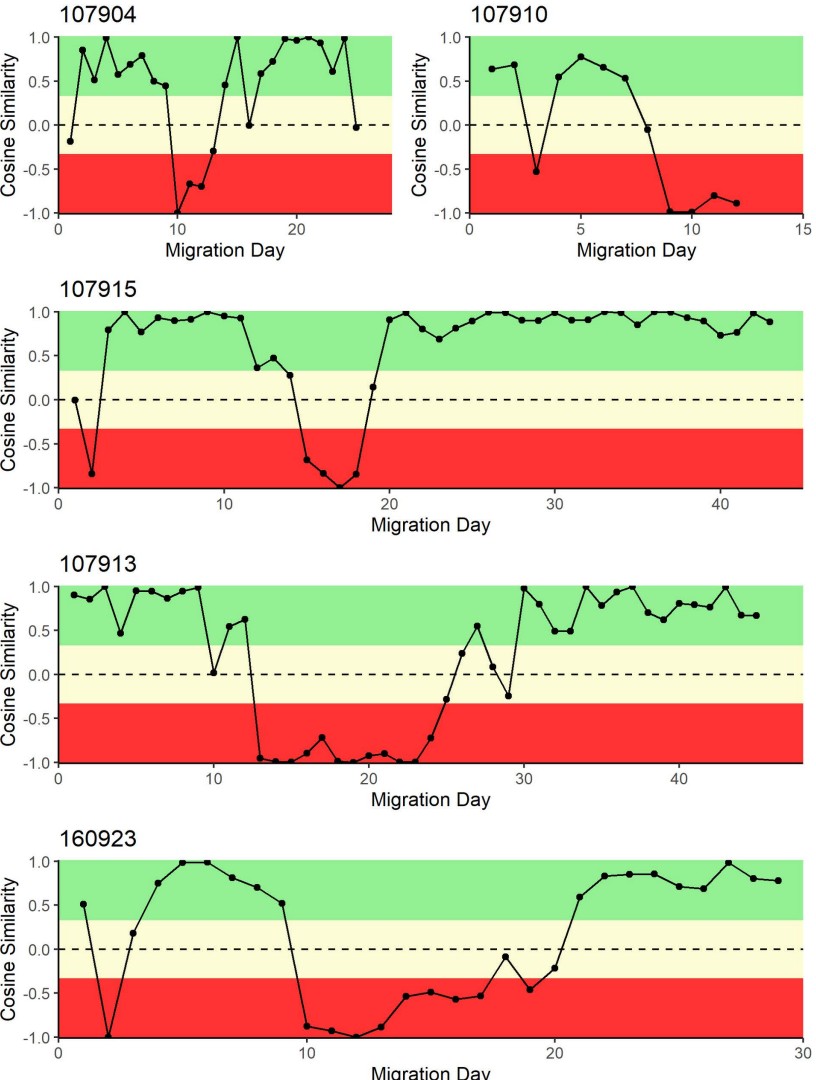

**Fig 11. Daily cosine similarity between turtle migration direction and ocean current direction for turtles tagged on Gandoca-Manzanillo National Wildlife Refuge, Costa Rica, and Chiriqui Beach, Panama (160923).** Cosine similarity close to +1 indicates that the direction of turtle movement was with ocean currents, whereas a cosine similarity close to −1 indicates a turtle movement direction against ocean currents.

## Post-nesting migrations

Post-nesting hawksbills in previous studies from the Caribbean basin have demonstrated two main strategies of migration back to foraging sites, with some migrating internationally to arrive at foraging sites long distances (respectively defined according to the study) from the nesting grounds [14,19,31–35,59–61], while others remained nationally in areas proximal to their nesting grounds [14,19,31–33,60]. Interestingly, the majority of turtles migrating from the insular Caribbean head west, southwest, or less frequently northwest (although see van Dam, Diez [19], Moncada, Hawkes [31], Hart, Iverson [35]).

All turtles in our study undertook regional, although relatively short-distance migrations, and all utilized migration corridors along the far Western Caribbean, making use of the relatively shallow continental shelf zones from Panama up to the peninsula of Mexico. Even

regional, relatively short-distance migrations require large amounts of metabolic energy, potentially impacting reproductive outputs and reducing the amount of foraging time available upon reaching the foraging site. However, there may be advantages (and presumably selective pressures) to maintaining fidelity to distant foraging grounds. Shimada et al. [62] suggest prior knowledge of the foraging area and an ability to survive long-term are definite benefits, likely to outweigh haphazardly establishing new foraging sites that may indeed have adequate food availability, yet whose long-term survivability is unknown. Additionally, the unknown factor of encountering predators in a new foraging habitat may reduce the suitability of the habitat [63], especially for turtles that have already experienced years of site suitability at their existing foraging grounds.

Coastal migrations are not always the most advantageous choice for migrating when water currents are strong. In our study, turtles nesting south of central Costa Rica instead chose longer routes that took advantage of circular current movements. This involved either being swept out by the Colombia-Panama Gyre, or actively swimming with the currents of the Gyre, then diverting away from it towards the coast, terminating somewhere along the Nicaraguan or north Costa Rican coasts. This route may mimic the path turtles took as hatchlings when they dispersed from their natal beaches for the first time and were swept out into ocean currents [64]. This idea may shed light on how they initially found their adult foraging habitats, since there is ample experimental and observational evidence demonstrating the ability of sea turtles to detect and orient to magnetic fields throughout their pelagic [65–67], neritic [68,69], and adult [70,71] stages. Nevertheless, more research on this aspect of foraging-ground finding is needed. In any case, adult turtles exited the Gyre at several different places, with some turtles returning to the shoreline in Nicaragua and swimming north, while another turtle (107913) remained in the Gyre back to central Costa Rica before turning north. Eventually, these turtles moved into foraging areas along Nicaragua and Honduras, most notably the Nicaragua Rise.

Our findings support Maurer and Eckert [30] that there is mounting evidence that the region off the northeast coast of Honduras along the Nicaragua Rise may well be a regional foraging "hotspot." This area has been highlighted in prior studies as an area that is important not only for post-nesting hawksbill turtles, but also green (*C. mydas*) [72] and loggerhead (*C. caretta*) [73] turtles as foraging grounds [74]. Further, this foraging area is important for hawksbills from throughout the Caribbean [14,18,31,33–35]. Becking et al. [61] found post-nesting turtles from Bonaire traveled as much as 1,766 km (*C. caretta*) and 3,135 km (*E. imbricata*) to reach this foraging area, while Maurer et al. [32] found post-nesting hawksbills from Jamaica traveled more than 723 km to reach and settle in the Nicaragua Rise area off the coast of Honduras. Given the paucity of ecological information available for this location, evidence from migrating nesting turtles in our study support the need for intensive investigations into habitat features, fishing pressures, resident population dynamics, and movement strategies of turtles in this potentially important foraging residency area.

One hawksbill from Costa Rica (107915) continued its migration along the continental shelf of Honduras to an area of open water approximately 50 km southeast of Utila where it remained for more than 700 days before moving west to an area just 3.5 km off the coast of Punta Izopo National Park, where it was tracked for an additional 400 days. This behavior is similar to that reported by Maurer et al. [32] of a post-nesting hawksbill from Jamaica undertaking migrations to two separate locations, remaining in the first for approximately 71 days before moving on to the second 403 km further south. In contrast, in our study turtle 107915 showed extremely long residence times in both locations. We are unsure why this turtle took up such long residence times in two different locations only approximately 52 km apart, and there is little available information regarding the health and suitability of reef habitats and

abundances of prey species for hawksbills in these locations. However, the second site where this turtle remained is within the marine area of the Jeannette Kawas National Park (JKNP), and therefore may provide some level of protection to turtles that forage there, although only recommendations have been proposed to fully protect the area of the park [75] and, to our knowledge, no legislation to protect and patrol the park since that time has been forthcoming. If the area of the JKNP has not moved beyond the status of a 'paper park,' turtles who forage in the area may be threatened by illegal direct take, as they are in other areas of coastal Caribbean Honduras.

It is of interest that none of the turtles tracked from PHB navigated southeast to forage in the same area as other adults foraging in the Cordelia Banks/Cayos Cochinos area, or northeast in the Nicaragua Rise hotspot. Instead, all of them moved west or northwest along the continental shelf, hugging the coastal zone. The entire coastline from the Yucatan through the Gulf of Honduras appears to have reserve areas important as foraging grounds for migrating hawksbills from coastal Central America, as well as the insular Caribbean [30,32,61], demonstrating the necessity of protected zones that may help mitigate the effects of legal and illegal take, inshore fisheries, and large-scale fisheries bycatch.

## Foraging home ranges

We found hawksbills making post-nesting migrations to foraging grounds had very small foraging habitats. These data infer that small area use may indicate high-quality habitats or food resources, sufficient resting sites, and stable temperatures that do not require turtles to move to thermoregulate. We tracked turtles within foraging habitats over multiple years; therefore, our results suggest they do not need to seek out new foraging grounds in order to acquire sufficient food resources. With the exception of one turtle from Costa Rica (107915) who established two different foraging home ranges with core UDs of 23 $km^2$ and 27 $km^2$, respectively (possibly due to insufficient prey abundances), the foraging home ranges turtles settled into appear to contain sufficient foraging resources. For Turtle 107915, we presume the primary foraging area had insufficient prey abundances to meet the energy needs of a single nesting female over the long term. Just to the north of this area, studies by Baumbach et al. [76] and Wright et al. [77] have demonstrated juvenile populations of hawksbills have small home ranges (<1 $km^2$) and suggest these small home ranges are sufficient for juvenile growth because of the high relative abundances of food species, including sponges (especially *Geodia neptuni*) and the red algae *Kalymenia leminguii*. For the other turtles we tracked, our core UD foraging home range data support the findings of other studies that have demonstrated small foraging home ranges of 2–50 $km^2$ [59,78,79].

The foraging UDs we report in the current study are different than some hawksbills tracked from the Dominican Republic by Hawkes et al. [14], who found turtles settled in foraging home ranges (using α-hulls) of from >1,000–4,422 $km^2$, in deep waters (>500 m) located from 4–195 km off the coast. Although some turtles in the current study migrated to habitats ~ 180 km off the coast (i.e., the Nicaragua Rise), they nevertheless maintained home ranges a fraction of the size of those for turtles from the Dominican Republic.

## Currents

Few studies have investigated the direction of travel of hawksbill turtles during post-nesting migrations in relation to surface currents. In the current study, we report detailed swim direction of hawksbills as daily $S_C$ plots for turtles and current directions. Our results demonstrate that most turtles from Honduras and Tortuguero, Costa Rica began their migrations either cutting across prevailing currents, or swimming directly against them. This differs from most

turtles tracked from GMNWR, Costa Rica, and Panama, who began their migrations by following the currents out into the ocean, then shifting between moving with and moving against currents multiple times through the remainder of their migrations. Additionally, turtles from the two more northern nesting grounds appeared to have fewer large changes in $S_C$, representing moving with versus moving against currents, than did turtles from the more southern nesting grounds. This, however, may be a function of the much longer migration lengths for turtles from both GMNWR and Panama compared with those from Honduras and TNP. The movements of turtle 107910 from GMNWR, whose shorter tracking duration was similar to those from the northern beaches, supports this hypothesis, as she demonstrated fewer shifts between moving with versus against the current.

Despite differences in nesting beaches, foraging habitats, and year of migration, all hawksbill turtles that we tracked displayed post-nesting migration directions that averaged close to a weighted $S_C$ of zero. Each turtle accomplished this via a different movement strategy, whether it involved traveling across currents for the entire migration, or dramatic zigzagging that changed relative direction every day (Figs 9–11). The largest variations to this pattern were turtles 108759 and 108760 that seem to have only swum with ocean currents. Another point of interest was that the re-migration of turtle 137659 involved traveling back to Tortuguero from her foraging habitat. In this case, the re-migration corridor was almost the post-nesting corridor in reverse; however, the weighted $S_C$ was much closer to zero than expected. We are unable, therefore, to assume that this turtle traveled with currents in one direction and against currents in the other direction. Instead, she was able to energetically mediate her efforts in regard to ocean movements in both the post-nesting migration and re-migration back to the nesting beach.

Our interpretations of migratory movements in relation to currents agrees well with reports by Horrocks et al. [59] of post-nesting hawksbills leaving Barbados cutting across southeasterly currents to avoid running headlong into northwesterly currents as the turtle was migrating northwest to Dominica. Additionally, those authors report another turtle moving southwest from Barbados cutting across both southeastern and northeastern currents, and ranging far west of the target foraging grounds in Trinidad, presumably allowing the turtle to avoid strong counter-currents for much of the migration south [59]. We agree that although turtles are fully capable of swimming against prevailing currents and tides over long distances [80], cross-cutting and utilizing current flows in the direction of migration are likely to reduce demands on depleted energy stores following months of mating and nesting activities. Taking advantage of prevailing currents may serve to minimize energy expenditure and travel times required to reach foraging sites [59]. This is further supported through work by Mencacci et al. [81], who tracked post-nesting female loggerhead turtles (*C. caretta*) displaced for the purpose of the study, in relation to oceanic sea surface temperature (SST) and geostrophic velocities. Those authors found turtles generally moved in the direction of prevailing currents, with few instances in which turtles clearly moved against currents, and where migratory routes were heavily influenced by oceanic features turtles encountered. Further, they found turtles appeared to actively swim in the same direction of currents, rather than passively drift along with prevailing currents [81]. Actively swimming in the same direction as currents, even during intermittent periods of long-distance migrations, is likely to increase swimming rates and reduce the overall duration of migrations between nesting and foraging grounds. This may explain some of what Hays et al. [82] found when they recorded turtles migrating towards islands based on course headings. While their subsequent model suggested that turtles were not being carried off course by currents, it would be interesting to test whether energetically canceling out ocean current movement is involved in the apparent course navigational system of sea turtles.

## Conservation implications

The importance of enacting multi-national agreements to protect sea turtles from direct take as well as fisheries bycatch along their migration routes is supported by our data, adding to the data collected in the greater Caribbean region [14,19,30–35,59–61]. Our study further stresses that foraging hotspots, such as those in the Nicaragua Rise, the Gorda Banks off Honduras' northeast coast, and the Cayos Miskitos off northern Nicaragua, require further investigation to better understand the importance of these habitats as foraging grounds for migrating turtles. Ongoing threats, such as direct legal take [83,84], and large- and small-scale fisheries bycatch [83–85] exist for the turtles in this study, who maintain migration routes along the shallow Inter-American coastal shelf. While these threats may, to some degree, be mitigated by the presence of established marine protected areas along the Central American coastline and Meso-American Barrier Reef, a large percentage of these are poorly funded, are under-managed, and have little evidence demonstrating their effectiveness in protecting endangered marine species [33,86–88].

The efficacy of the protection provided to turtles by MPAs will depend on international cooperation and intentional effort to mitigate the effects of both legal and illegal fisheries found in sea turtle migration corridors. These efforts, in turn, will need to be supported by data elucidating the small-scale, high-resolution movement of turtles within and outside of these protected zones. We suggest there is a need for continued satellite tracking of nesting hawksbill turtles from locations throughout Central America and the insular Caribbean with higher resolution/accuracy locations to fill the gaps in our knowledge regarding hawksbill nesting beaches, migration corridors, and foraging grounds in the Western Atlantic.

## Acknowledgements

We would like to extend special thanks to ProTECTOR, Inc. Country Director, Lidia Salinas for securing research permits in Honduras, Jimmy Miller for logistical support on Roatan, Chase Kent-Dotson for his assistance with data analyses, and two anonymous reviewers whose comments and suggestions helped to improve the presentation of the manuscript. This is Contribution No.47 of the Marine Research Group (LLU) and Contribution No.25 of ProTECTOR, Inc.

## Author contributions

**Conceptualization:** Stephen G. Dunbar, Daniel R. Evans, Lindsey R. Eggers, Quintin D. Bergman, Frank V. Paladino, Lidia Salinas, Chelsea E. Durr.

**Data curation:** Stephen G. Dunbar, Daniel R. Evans, Lindsey R. Eggers, Quintin D. Bergman, Luis G. Fonseca, Frank V. Paladino, Chelsea E. Durr.

**Formal analysis:** Lindsey R. Eggers, Chelsea E. Durr.

**Investigation:** Stephen G. Dunbar, Daniel R. Evans, Lindsey R. Eggers, Quintin D. Bergman, Frank V. Paladino, Chelsea E. Durr.

**Methodology:** Stephen G. Dunbar, Daniel R. Evans, Lindsey R. Eggers, Luis G. Fonseca, Frank V. Paladino, Lidia Salinas, Chelsea E. Durr.

**Project administration:** Stephen G. Dunbar, Daniel R. Evans, Frank V. Paladino, Lidia Salinas, Chelsea E. Durr.

**Resources:** Stephen G. Dunbar.

**Software:** Chelsea E. Durr.

**Supervision:** Stephen G. Dunbar, Frank V. Paladino, Lidia Salinas.

**Validation:** Stephen G. Dunbar, Daniel R. Evans, Lindsey R. Eggers, Quintin D. Bergman, Chelsea E. Durr.

**Visualization:** Chelsea E. Durr.

**Writing – original draft:** Stephen G. Dunbar, Daniel R. Evans, Lindsey R. Eggers, Chelsea E. Durr.

**Writing – review & editing:** Stephen G. Dunbar, Daniel R. Evans, Lindsey R. Eggers, Quintin D. Bergman, Luis G. Fonseca, Frank V. Paladino, Lidia Salinas, Chelsea E. Durr.

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
