## [Decision Letter · Decision Letter 0]

11 Nov 2024

PONE-D-24-40236Inter-Nesting Area Use, Migratory Routes, and Foraging Grounds for Hawksbill Turtles (Eretmochelys imbricata) in the Western CaribbeanPLOS ONE

Dear Dr. Dunbar,

Thank you for submitting your manuscript to PLOS ONE. After careful consideration, we feel that it has merit but does not fully meet PLOS ONE’s publication criteria as it currently stands. Therefore, we invite you to submit a revised version of the manuscript that addresses the points raised during the review process.

Thank you for your patience during the review process. We have now received feedback from two reviewers. One reviewer is well-versed in Hawksbill turtles, while the other has extensive knowledge of satellite tracking in sea turtles. Both reviewers responded positively to the manuscript, highlighting that the study provides valuable insights into the species' biology. I also found the study straightforward, with data clearly presented.

However, some values in the manuscript lack associated error terms, and certain error terms require clarification regarding their type (e.g., standard error, standard deviation, confidence interval, or range). Please ensure that error terms are included for all average values and specify their nature. This applies particularly to values presented in the tables. Additionally, the tables appear to be cut off on the right side; please verify that they are formatted correctly. There may also be some issues with the figures, as noted by one of the reviewers.

Finally, please do not forget to review comments made directly to the manuscript (see attached document). 

We look forward to receiving your revised manuscript.

Kind regards,

Masami Fujiwara, PhD

Academic Editor

PLOS ONE

Journal requirements:    When submitting your revision, we need you to address these additional requirements. 1. Please ensure that your manuscript meets PLOS ONE's style requirements, including those for file naming. The PLOS ONE style templates can be found at https://journals.plos.org/plosone/s/file?id=wjVg/PLOSOne_formatting_sample_main_body.pdf and https://journals.plos.org/plosone/s/file?id=ba62/PLOSOne_formatting_sample_title_authors_affiliations.pdf 2. We note that you have indicated that there are restrictions to data sharing for this study. PLOS only allows data to be available upon request if there are legal or ethical restrictions on sharing data publicly. For more information on unacceptable data access restrictions, please see http://journals.plos.org/plosone/s/data-availability#loc-unacceptable-data-access-restrictions.  Before we proceed with your manuscript, please address the following prompts: a) If there are ethical or legal restrictions on sharing a de-identified data set, please explain them in detail (e.g., data contain potentially identifying or sensitive patient information, data are owned by a third-party organization, etc.) and who has imposed them (e.g., a Research Ethics Committee or Institutional Review Board, etc.). Please also provide contact information for a data access committee, ethics committee, or other institutional body to which data requests may be sent. b) If there are no restrictions, please upload the minimal anonymized data set necessary to replicate your study findings to a stable, public repository and provide us with the relevant URLs, DOIs, or accession numbers. For a list of recommended repositories, please seehttps://journals.plos.org/plosone/s/recommended-repositories. You also have the option of uploading the data as Supporting Information files, but we would recommend depositing data directly to a data repository if possible. We will update your Data Availability statement on your behalf to reflect the information you provide.

Reviewers' comments:

Reviewer's Responses to Questions

**Comments to the Author**

1. Is the manuscript technically sound, and do the data support the conclusions?

Reviewer #1: Yes

Reviewer #2: Yes

2. Has the statistical analysis been performed appropriately and rigorously? 

Reviewer #1: Yes

Reviewer #2: Yes

3. Have the authors made all data underlying the findings in their manuscript fully available?

Reviewer #1: Yes

Reviewer #2: Yes

4. Is the manuscript presented in an intelligible fashion and written in standard English?

Reviewer #1: Yes

Reviewer #2: Yes

5. Review Comments to the Author

Reviewer #1: This paper provides valuable insight into the inter-nesting, and foraging areas, and migratory routes of hawksbill turtles in a region for which there is a definite lack of information. I conducted satellite tagging of loggerhead and greens turtles for a number of years and fully aware of the effort and field conditions involved. I hope the data gathered in this project will assist resource managers in better defining areas that need additional protection from the impacts of commercial fishing activities, both legal and illegal. The paper is well-written and with a few minor editorial changes is good to go.

Reviewer #2: Overall a well written, clean MS. I have no comments that would substantively change the MS that would not be a matter of personal preference only.

The figs attached to the end of the document are in the wrong order - if this is published make sure they are arranged in the order you want.

6. PLOS authors have the option to publish the peer review history of their article (what does this mean? ). If published, this will include your full peer review and any attached files.

**Do you want your identity to be public for this peer review?** For information about this choice, including consent withdrawal, please see our Privacy Policy .

Reviewer #1: **Yes: ** David S. Addison, Senior Biologist, Conservancy of Southwest Florida, Naples, FL

Reviewer #2: No

---

## [Author Response · Author response to Decision Letter 1]

26 Dec 2024

We have provided a full Responses to Reviewers document in the re-submission of the revised ms in which we detail all responses to comments made by Reviewers and the Editor.

---

## [Editor Report · Decision Letter 1]

5 Jan 2025

Inter-Nesting Area Use, Migratory Routes, and Foraging Grounds for Hawksbill Turtles (Eretmochelys imbricata) in the Western Caribbean

PONE-D-24-40236R1

Dear Dr. Dunbar,

We’re pleased to inform you that your manuscript has been judged scientifically suitable for publication and will be formally accepted for publication once it meets all outstanding technical requirements.

Kind regards,

Masami Fujiwara, PhD

Academic Editor

PLOS ONE
---

## [Editor Report · Acceptance letter]

PONE-D-24-40236R1

PLOS ONE

Dear Dr. Dunbar,

I'm pleased to inform you that your manuscript has been deemed suitable for publication in PLOS ONE. Congratulations! Your manuscript is now being handed over to our production team.

Kind regards,

on behalf of

Dr. Masami Fujiwara

Academic Editor

PLOS ONE